# "It's like a book in the palm of my hand": Adapting the Safe Delivery App for Papua New Guinea to improve quality of maternal and newborn care

Delly Babona[1], Lucy Au[2], Cherolyn Polomon[3], Arpita Deb[4], Hilde Cortier[4], Disha Agarwalla[4], John Bolnga[5,6], Michaela A. Riddell [7,6], Angela Kelly-Hanku [7,6], Caroline S. E. Homer[1], Lisa M. Vallely [7]*

1 The Burnet Institute, Melbourne, Australia, 2 UNFPA PNG, National Capital District, Port Moresby, Papua New Guinea, 3 Pacific Adventist University, Koiari Park Campus, Port Moresby, Papua New Guinea, 4 Maternity Foundation, Copenhagen, Denmark, 5 Madang Provincial Health Authority, Madang Province, Papua New Guinea, 6 Papua New Guinea Institute of Medical Research, Goroka, Eastern Highlands Province, Papua New Guinea, 7 The Kirby Institute, UNSW, Sydney, Australia

* lvallely@kirby.unsw.edu.au

## Abstract

### Background

Health workers in many low- and middle-income countries are not adequately trained to provide quality antenatal and intrapartum care. The freely available Safe Delivery App (the App) provides health care professionals with direct and instant access to evidence-based, up-to-date clinical guidelines equipping them with an on-the-job reference guide, even in the most remote areas. In this paper we describe the uptake and acceptability and the process to align the App in Papua New Guinea (PNG).

### Methods

Between June 2022 and December 2024, quantitative and qualitative methods were used to explore the usage and acceptability of the App among health care workers in PNG, re-designing images and aligning clinical content with PNG national clinical guidelines. Dissemination of the App took place through formal and informal networks, including a training of trainers' program in one province.

### Results

The App was seen as an acceptable and useful tool among midwives and nurses working clinically, particularly those in remote areas and midwifery educators. There was an increase in the use of the App, from 354 registered users in 2022 to 1304 in 2024. The majority of users were midwives and nurses, working in primary health care facilities and nursing and midwifery establishments. Participants reported that

**Data availability statement:** Safe Delivery App user data: The data used in this study were collected through the Safe Delivery App and stored on cloud servers hosted by the Maternity Foundation in India and Central Europe. Due to privacy concerns and user confidentiality agreements, the raw user-level data cannot be shared publicly. However, aggregated and anonymised summary statistics that support the findings of this study are available from the Maternity Foundation upon reasonable request. Researchers seeking access may contact the Maternity Foundation at mail@maternity.dk and must comply with applicable ethical and data privacy regulations." The code book for the qualitative data is included as a supplementary file. Additional requests for qualitative data may be forwarded to info@pngimr.org.pg.

**Funding:** This study was made possible through philanthropic donations from the Finkel Foundation and a Burnet Institute Christmas Appeal, secured by CH and LV. The funders had no role in study design, data collection and analysis, decision to publish, or preparation of the manuscript.

**Competing interests:** The authors have declared that no competing interests exist.

the App has led to changes in practice, supporting and encouraging staff to follow evidence-based guidelines, improving their clinical management.

## Conclusion

The Safe Delivery App is seen as a useful tool, supporting clinical practice, knowledge and skills, providing users with more confidence in their ability to provide quality maternal and newborn health care. Wider implementation of the App across PNG may be a potential way to support health care workers in the remote settings, providing up to date evidence based clinical guidance in the absence of skilled midwives and doctors.

## Introduction

Despite gains in reducing maternal mortality, progress remains slow [1]. More than 90% of maternal and newborn deaths and 84% stillbirths occur in low-and middle-income countries (LMICs), the majority could be avoided with quality care during pregnancy and birth [2,3].

In response to the high burden of maternal and newborn deaths in LMICs, the Maternity Foundation [4], in collaboration with Universities of Copenhagen and Southern Denmark, developed and launched the Safe Delivery App [5] in 2015. The Safe Delivery App, hereon referred to as 'the App", is a freely available application that provides health care professionals with direct and instant access to evidence-based, up-to-date clinical guidelines. The App covers 17 topics on Basic Emergency Obstetric and Newborn Care and critical maternal and newborn health topics [5] through five key features: (i) animated videos, (ii) instructional action cards, (iii) practical procedures, (iv) drugs list and (v) "My-Learning". The MyLearning platform allows for competency-based learning and training with users able to test their knowledge interactively at their own time and pace. On completion of all topics, formal certification is awarded to become a Safe Delivery App Champion.

The aim of the App is to improve knowledge and skills among health care professionals, improving the quality of life-saving care. It provides the opportunity to equip birth attendants, even in the most remote areas, with an on-the-job reference guide. Importantly, once downloaded, the App does not require internet connection to function.

The App is currently used by more than 455,000 health care professionals in over 100 countries in Sub-Saharan Africa, the Middle East and Northern Africa, Latin America and the Caribbean, Asia and the Pacific. It has been shown to improve outcomes for women and their newborns in LMIC settings [6–11]. The App is available in five Global language versions (English, Spanish, French, Portuguese and Arabic), has been adapted in more than 20 countries and translated into more than 30 local languages.

Papua New Guinea (PNG), a low-middle-income country in the Pacific region is geographically, socially and linguistically diverse. The majority (87%) of the 10.5

million people live in rural areas with poor road networks, infrastructure and access to health care. PNG has high maternal and neonatal mortality rates with estimates of 171–584 per 100,000 and 20–29 per 1,000 live births respectively, compared to corresponding global figures of 223 and 18 [12–14].

With a critical shortage of all cadres of health care workers, particularly midwives [15], most maternal and newborn care is provided by nurses or community health workers (2-year formal training program). Often working in rural locations, with limited infrastructure, many health care workers are required to manage maternal emergencies and sick newborn infants alone, with limited support. Access to ongoing education, especially up-to-date and current information is a challenge [16], with limited access to regular education on emergency care. The App can provide up-to-date information to maternity care providers, especially in rural and remote settings. The use of digital technology in PNG is growing; around 50% of the population own a mobile phone with almost universal ownership among health care workers [17].

The App was introduced in PNG in October 2020 through the United Nations Population Fund (UNFPA) PNG in partnership with the National Department of Health and Provincial Health Authorities. It was initially launched to support the COVID 19 response. Over one hundred (108) health care workers received instruction in the use of the App. Sixteen midwives from this cohort went on to be trained as trainers of the App (July 2021), supported by Maternity Foundation and UNFPA PNG. This was the first time that the App had been used in the Pacific region.

By October 2021, the App had been downloaded 378 times in five provinces in PNG (Western, East New Britain, Eastern Highlands, Southern Highlands and Central). Most of the users were midwives (58%), with 40% of all users working in primary care facilities. Half of the users had learnt about the App through in-service training, others had received stand-alone training in the use of the App, learnt about it through a colleague or friend or through a conference or other event.

Despite these downloads in PNG, it is not known how the App has been received in the country, and if its use could be strengthened in PNG, or also implemented in other Pacific Island Countries. The overall aims of this study are to (i) describe the usage and acceptability of the App in PNG; and (ii) describe the process to develop the App in PNG; adapting the App for the PNG context and conducting a hybrid model of training for midwives and nurse and midwifery educators (East New Britain).

## Methodology

A mixed methods study was conducted in three phases between 30 June 2022 and 31 December 2024 (Fig 1). In Phase 1, the usage and acceptability of the App among health care workers in PNG was explored. In Phase 2, the App was adapted to the PNG context, re-designing images and aligning clinical content with PNG national clinical guidelines. Phase 3 included two workshops providing a "deep dive" introduction to the App and a training of trainers' program. Participants included nurses, midwives, student nurses and midwives and nursing and midwifery educators (Table 1). Ongoing dissemination of the App took place through formal and informal networks throughout each of the phases.

Ethical approval was gained from the Papua New Guinea Institute of Medical Research (PNGIMR; IRB 2115); the PNG Medical Research Advisory Committee of the National Department of Health (MRAC 22.11) and UNSW Sydney's Health, Medical, Community and Social ethics committee (HC220246).

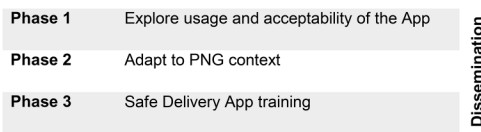

**Fig 1. Phases of the study.**

**Table 1. Participants included in different phases of the review.**

| In depth interviews | 7 |
|---|---|
| Midwifery Educators | 2 |
| Midwives | 4 |
| Obstetric unit manager/ midwife educator | 1 |
| **Focus group discussion** | **12** |
| Nurse lecturer | 1 |
| Student midwife | 5 |
| Midwives | 4 |
| Midwifery educator | 2 |
| **Deep dive orientation** | **31** |
| Student nurses | 11 |
| Student midwives | 7 |
| Nurses | 4 |
| Midwives | 4 |
| Nurse educator | 2 |
| Midwifery educator | 3 |
| **Training of trainers** | **12** |
| Midwife | 6 |
| Nurse educator | 3 |
| Midwifery educator | 3 |

## Phase 1: Usage and acceptability of the App

Quantifiable de-identified data were collected at baseline and in December of each year (2022−24). Data included the number of users, level of training, years in clinical practice, work location and features and topics most frequently accessed. These data were provided by Maternity Foundation, as part of their ongoing monitoring of the use of the App. Users of the App consented to the sharing of their de-identified data when registering their user profile within the App.

Qualitative data were collected between 30th June and 17th September 2022. Health care workers with experience in using the App were purposively identified with the support of UNFPA. Inclusion criteria included health care workers who had received training in the use of the App and were currently using or had used the App since their training. Interviews were conducted in English, guided by a semi-structured interview guide. Individual interviews were conducted with seven health care professionals and one focus group discussion with 12 midwifery and nurse educators and student midwives (Table 1). Interviews were conducted either via WhatsApp, Zoom or face to face. For interviews conducted face to face, written informed consent was obtained; for those conducted virtually we used a consent script to gain verbal consent.

All interviews and the focus group discussion were digitally audio-recorded. Interviews were conducted in English by one of two investigators (LV, LA) and took between 30–40 minutes. The focus group took 45 minutes and was conducted by two investigators (LV, CH). All interviews were transcribed by trained members of the social science team at the PNGIMR. Interviews were de-identified, and a pseudonym assigned prior to transferring and storing into a password protected computer at the PNGIMR.

All interviews were coded using NVivo qualitative software V12 (QSR International). A coding framework was developed (Supp file 1) with a focus on experiences of using the App as well as exploring perspectives and ideas of how it could be adapted for future use in PNG. The transcripts were cross-checked for consistency and revised using inductive and deductive thematic analysis, a process whereby transcripts are read for over-arching and sub-themes and re-read to identify intersections between codes, differences, and similarities within and across transcripts (S1 File ).

## Phase 2: Adaptation of the Safe Delivery App

Review and re-design of the illustrations in the App took place between June 2022 and January 2023. Working with Maternity Foundation, UNFPA PNG and investigators, images were re-designed to depict women and health workers in PNG. Between October and November 2022, review of clinical content was conducted by lead investigators (DB, JB, LV). This led to a two-day workshop in Port Moresby, PNG (November 2022) to review clinical content in the App alongside PNG guidelines, guided by obstetricians, paediatricians and midwifery educators. Clinical content was reviewed and adapted to the local context ensuring alignment with the current Obstetrics and Gynaecology and Paediatric clinical guidelines [18,19].

## Phase 3: Safe Delivery App training

In October 2024 two workshops were conducted in East New Britain Province, PNG: a one day "deep dive" training program and a 4-day training of trainers' program. Each group included midwives, nurses, midwifery and nursing students and midwifery educators. Training was led by clinical trainers from Maternity Foundation, who were present online and two co-facilitators onsite to facilitate in-person and provide the participants hands on practice sessions (DB, LV) through simulated skills-based learning platforms.

The deep dive training included 31 participants who received a general orientation to the App, including how to install the App, register and create their identity in the MyLearning platform. The five features were explained through a combination of lectures and activity-based lessons, followed by exercises using the normal labour and birth topics within the App.

The training of trainers included 12 participants, nine of whom were already familiar with the App through the earlier introduction in 2021. Theoretical sessions were conducted online by the clinical trainers from Maternity Foundation, with the skills activities and assessments facilitated by the onsite facilitators (DB, LV). Practical activities focused on three modules: active management of third stage of labour, postpartum haemorrhage (PPH) and manual removal of the placenta. The overall aim of the training of trainers was to enable them to cascade the training to other health care workers in PNG.

Throughout the study period, the App was discussed and disseminated in several forums, including through the PNG Midwifery Society, the PNG Midwifery community of practice WhatsApp group (which has 751 members – midwives, nurses, doctors and community health workers), and one non-government organisation in East Sepik Province, PNG.

## Results

### Phase 1: Usage and acceptability

At the start of the study period there had been 378 downloads of the App. Between June 2022 and December 2024 there were an additional 1495 downloads (Fig 2); 87% of users (1304/1495) had registered profiles, providing detail about use of the App. The majority were midwives (45%; 591/1304) and nurses (32%; 414/1304); and 11% (146/1304) were students (nurses and midwives); there was an increase in use by nurses over the study period (Fig 3). About one third (31%; 399/1304) of users worked in primary health care facilities; 23% (297/1304) worked in nursing and midwifery education establishments. There was a slight increase in use at primary health care facilities during the study period from 27% (94/354) to 29% (162/572) in December 2022 and 2024, respectively.

Most users had learnt about the App through a colleague or employer (32%; 412/1304) or in service training (28%; 362/1304). Among all users, 29% (381/1304) had more than 11 years clinical experience; 20% had between one- and five-years clinical experience and 26% (334/1304) were students nurses and student midwives (Fig 3).

Of the five key features included in the App, the use of "Action cards" was the most frequently accessed feature(89%; 1161/1304), followed by "MyLearning" (89%; 1162/1304) and "Video chapters" (82%; 1063/1304) (Fig 4). The most frequently accessed topics were normal labour and birth and hypertension with 67% (868/1304) and 62% (805/1304), respectively, of all users accessing these topics (Fig 4).

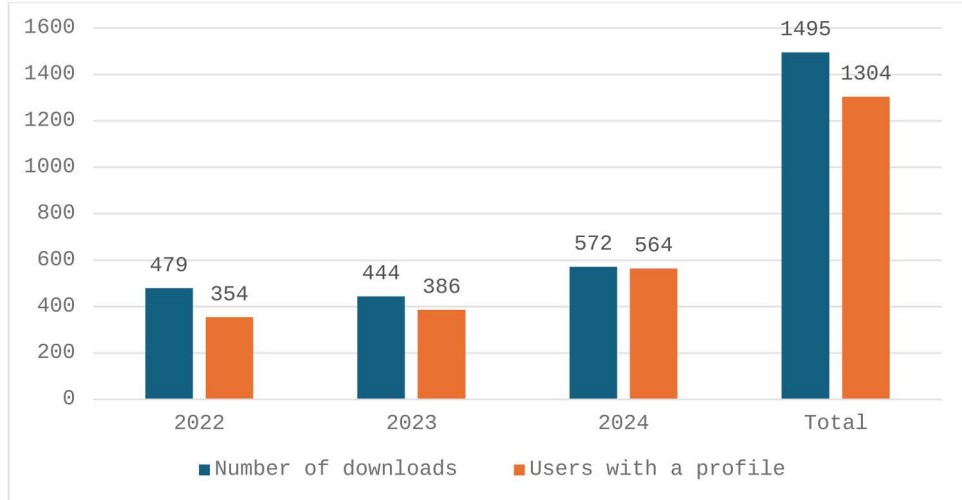

**Fig 2. Downloads and users with a profile.**

## Acceptability

The App was seen as an acceptable and useful tool among midwives and nurses working in the clinical setting as well as midwifery educators, particularly those remote areas. The ease of use as an on-the-spot reference tool was seen as valuable:

> … up in the Highlands; it's rural… this app it's really helpful… most of us working in rural, sometimes we work without doctors and HEOs and we [as a] midwife work alone….I worked as a nurse and [/but/] as a student midwife, I see that the app is like it really helps us. So, I wish to go back and share this with some of my colleague midwives in hospital.

Maria, Student midwife, Enga Province (FGD)

> I wish I had this application long time ago.

> This is very handy; it's like a book in the palm of my hand… I just carrying it anywhere.

Susan, Midwife Educator, East Sepik Province (FGD)

Most of the participants had received some training on the use of the App following the early introduction in 2020. Two midwives who had not received any training in its use had been introduced to the App through a contact at work and continued to use it throughout their work, finding it easy to navigate and use in their day-to-day work. In another setting, one midwife encouraged health care workers in rural centres to download the App with positive feedback.

> I have a network out at the rural communities, [and] they call me up if they are facing any complications. So, I simply asked them if they have an android phone and then I instructed them to install it [the app]…. the feedback they gave me was overwhelming [positive]… they got the updated information, and it was very helpful. And this app already saved one mother with retain placenta; with the community health worker, it's not a midwife that performed [the procedure]. ….

Lily, Midwife, Office in Charge, Morobe Province (FGD)

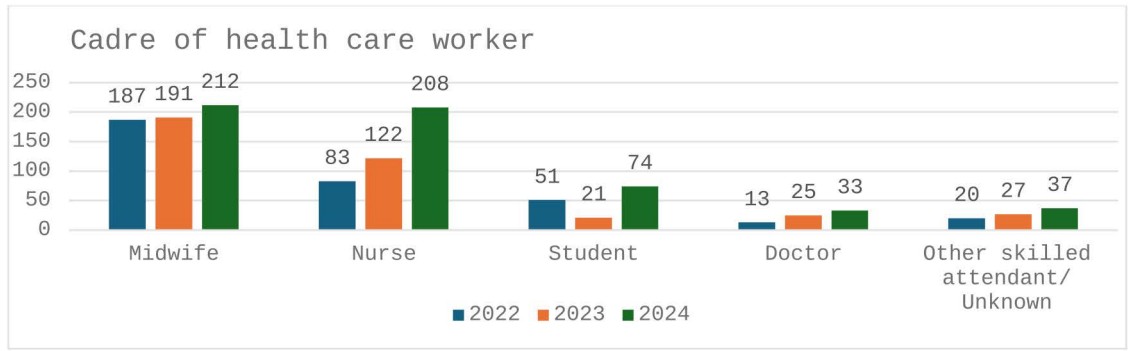

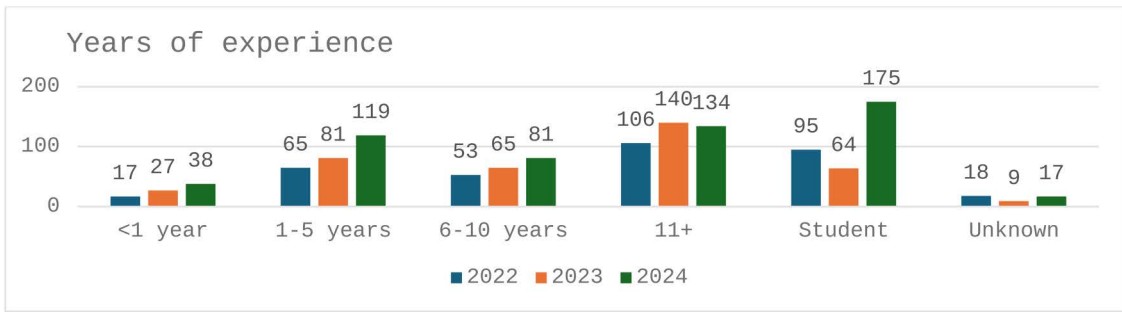

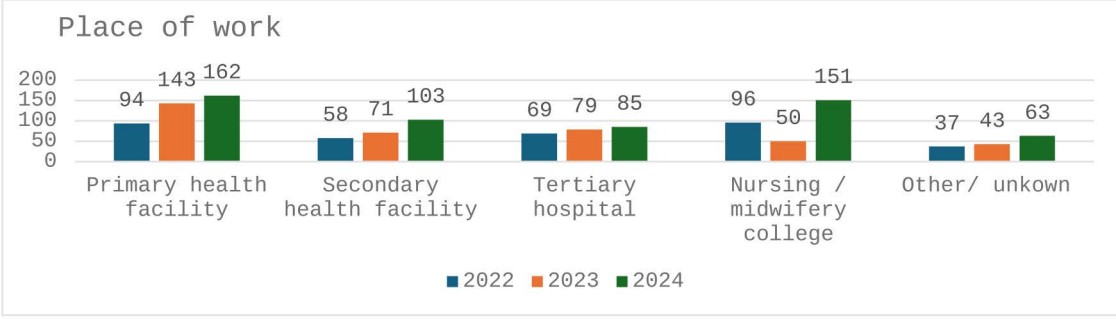

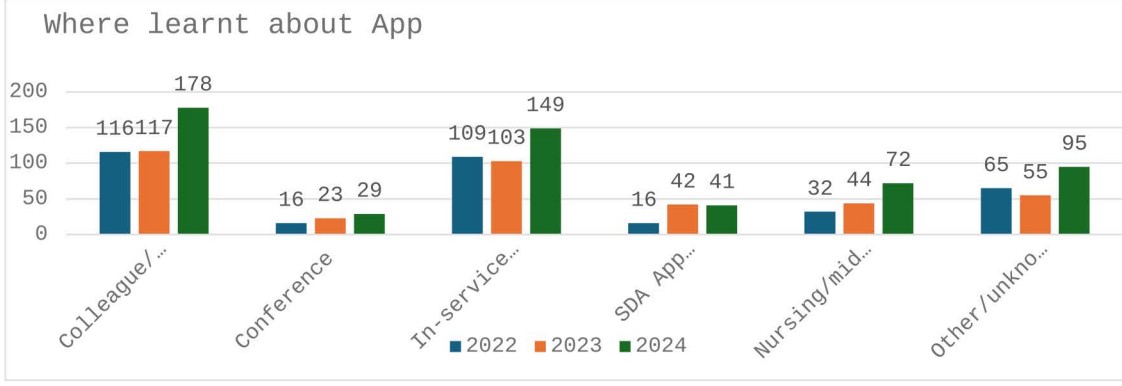

**Fig 3. Experience and place of work.**

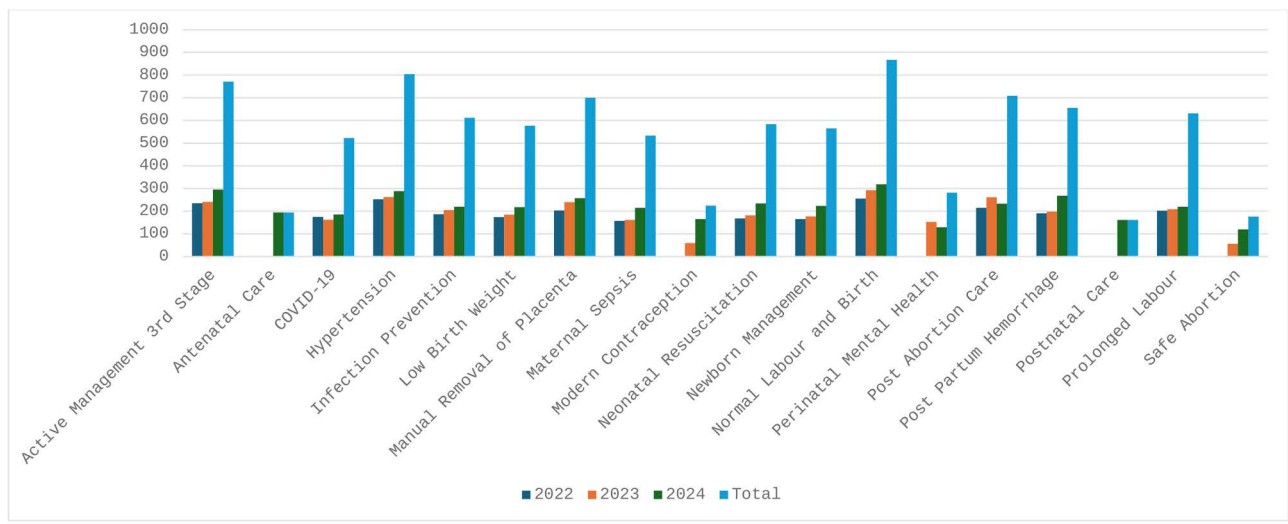

**Fig 4. Safe Delivery App usage.**

Referring to the different features of the App, one midwifery educator spoke of the importance and usefulness of the App, catering for different types of learners.

*…I will say [the features] are important. All of them are user-friendly… coming from an education background, we have different types of learners… there is a learner that will have to do something to follow, there is a learner that by just watching the video will already understand, there is another person that has to read and understand. So that, the whole app itself is so useful...*

Hilda, Midwife Educator, Eastern Highlands Province

## The App is supporting clinical care

The App has enabled health care workers in the health centres to manage situations without the need to refer women to hospital, with one participant reporting a reduced number of referrals. Another participant reported that after training, staff were able to use the App, leading to a reduced need to call support from senior staff.

*According to my statistics (data), I don't do manual removal of placenta now… because of the app and the active management of first stage of labour. Especially the bladder, (emptying the bladder with a urinary catheter) and the repeat of oxytocins…those are the two things my staff have learnt in active management of third stage of labour… they are putting into practice and that brings change in their performance...[they] no longer come [to the] centre now and I don't do the manual removal of placenta – they follow the protocol they have seen in the app".*

Jackie, Senior Midwife and Unit Manager, Autonomous Region of Bougainville

While designed as a reference tool used for those working in rural areas, with limited clinical support, one informant reported training all the staff working on a labour ward in the use of the App. Another reported referring to the App up to three times a week during busy antenatal clinics, to support the management of pre-eclampsia among pregnant women.

   

*According to my assessment, my staff can do appropriate hypertension and management of PPH. They can be able to fully manage PPH, that's including my HEO…in the unit, she improved a lot.*

Jackie, Senior Midwife and Unit Manager, Autonomous Region of Bougainville

In cascading their training, participants introduced the App to student midwives; others had used the App to provide refresher training for midwives, reinforcing and updating clinical practice.

*… the training that was given to us was for all the midwifery schools in the country, [then] we went back and then imparted the knowledge [to] our midwives. So, when midwives then go to the facility, they will train their own staff in the facility… this is transfer of information.*

Hilda, Midwife Educator, Eastern Highlands Province

Midwifery students used the App as a reference tool to support their learning. The continuation of the use of the App in ongoing practice was described by midwifery educators, including in their emergency obstetrics and newborn care training.

*…so students are able to differentiate the hypertension, preeclampsia [because of the App]. They are able to differentiate severe preeclampsia and eclampsia, and at least they know how to treat them. Because the directives are also there, how to use the magnesium sulphate…*

Philippa, Midwifery Lecturer, National Capital District

The App has led to changes in practice for some health care workers, supporting and encouraging health staff to follow evidence-based guidelines to improve their clinical management.

*Sometimes it is the community health workers… they come across mothers with pre-eclampsia so they just let me know that they have come across a case [and] want to give magnesium sulphate… I just tell them to—I usually tell them to go ahead and give it. Before that, they need supervision like a HEO or a midwife to stay with them before they give magnesium sulphate. Now they give it on their own.*

Jackie, Senior Midwife and Unit Manager, Autonomous Region of Bougainville (FGD)

A number of participants spoke of referring to national standard clinical guidelines for obstetric care, often referring to "the red book", a small, pocket sized book outlining key aspects of care during pregnancy and childbirth, aligned with the national standards. However, often these books are not available in rural areas, being provided primarily to midwives.

*I did find it very helpful… when we were coming across mothers who were being identified as pre-eclamptic or hypertensive mothers. Though we had the red obstetric handbook but.information which [was] in the app [was] really helpful when managing cases like this …*

Ben, Research Midwife, East New Britain

The most frequently mentioned topics used to support clinical practice were retained placenta, newborn resuscitation, hypertension and pre-eclampsia, post-partum haemorrhage and management of low-birth-weight infants. Video clips tended to be used the most, followed by action cards. One informant described the use of the video as not only being easier than reading through procedures but useful in a specific situation, for example, it could be played while following instructions to help the woman, reducing the delay in providing care:

*…we brought her to the health centre… there wasn't any senior midwife there, just me and [another nurse] and she wasn't experienced in giving magnesium sulphate as she was a junior CHW at that time….So I got out the app, I was watching it the whole time because I knew about the things of hypertension. So…we followed the procedures and I got [the nurse] to help me out.*

Mary, Research Midwife, Madang Province

Another informant described the videos as being easier to follow, rather than the action cards. He went onto to say that the action cards contained some clinical language that may not be understood, but in the video with the instruction alongside the video it could be more easily followed.

*… I'd say the language used in the videos, some of the terms that are used…they are clear. But the ones that are written in the action cards…the ones that are in writing, let's say I think some of the terms in there would be a bit hard to understand…*

Ben, Research Midwife, East New Britain

## Challenges with using the App

A few challenges with the App were identified, including issues with poor internet and network coverage for the downloads and updates. However, once the App was downloaded it worked fine, there was just concern that updates could potentially be missed.

One informant described some difficulty in using the App initially, especially navigating the registration and profile data. But once this had been achieved, they found the App to be easy and intuitive to use. There were concerns that the App may not be easily used by older workers. One informant highlighted the importance of needing an adequate battery life, a concern particularly in a rural area where electricity may not always be available.

While it is recognized that many people in PNG now have access to or own a mobile phone, some participants suggested that a mobile phone be available at the health facility for all staff to use. It was also felt that this could also be used to seek clinical support and arrange patient transfers where necessary.

*…most [of the] population would have access to a cell phone and especially an android so they are able to download the app so they are able to access it when they needed it. But the other approach…if we could …purchase a phone for the facility …[it] will have the app already there and it will be left in the health centre and any person attending to a labouring mum or antenatal mum would always have access to the phone at the facility.*

Hilda, Midwife Educator, Eastern Highlands Province

Overall, participants recommended that the App be available to be used more widely among all levels of health care workers in PNG. There was some discussion about improving the images and some concern of other key emergencies not included, for example:

*…when I was going through the app I realise that some of the emergency complication…I [would] like the steps in doing twin delivery, vacuum, breech presentation and other emergency procedures. That will be helpful for us…the remote midwives.*

Val, Student midwife, Autonomous Region of Bougainville (FGD)

There was also some discussion about translating the App into Tok Pisin (the local commonly used language), in particular the video clips:

*I prefer if we can also have it translated to the pidgin (Tok Pisin) language, so yeah it can be easy for those that who are in the remote area who are trying to help us.*

Michelle, Midwife, Central Province (FGD)

## Phase 2: Adaptation to PNG version of the App

Findings from interviews were used to inform refinement, enhancement, and adaptation of the App during Phase 2. All images were shared with the wider investigator team and all images finalised with the visual adaptation for PNG completed and available for download in in January 2023. At this time there were 13 topics available in the App.

Review of the clinical content was conducted for all 13 modules, with a focus on six topics reviewed in detail with the expert group. The six topics included managing normal labour and birth, low birth weight infants, resuscitation of newborns, prolonged labour, post-partum haemorrhage and hypertension. All required revisions were shared with Maternity Foundation and the first round of revisions were completed in July 2023. In December 2023 the second version of the PNG Safe Delivery App, aligned with national clinical standard guidelines was published and available for download in both Google play and App Store. Following the review and alignment with national clinical guidelines, an additional four new topics– antenatal care, postnatal care, perinatal mental health and modern contraception was made available in an updated version in December 2024 (Fig 5).

## Phase 3: Introduction to App and training of trainers

Among the 31 participants included in the deep dive training, all were able to install the App on their phones, the majority found it easy/ very easy to use and found it useful/very useful. The practical procedures followed by the videos were the most liked features of the App; all found MyLearning relevant; and the majority said they were extremely likely to recommend the App to colleagues. Overall, participants were satisfied or very satisfied with the training, and specifically with the duration of the training and the facilitators' knowledge and competency.

The 4-day training of trainers included simulated practice utilising skill stations. All participants reviewed and practiced clinical skills through skill stations, practicing active management of third stage of labour, bimanual compression, aortic compression, uterine balloon tamponade and manual removal of placenta. All 12 participants reached the expert level in the MyLearning platform for active management of third stage of labour, postpartum haemorrhage (PPH) and manual removal of placenta by the fourth day. An increase in knowledge levels was seen among all individuals who completed pre and post assessments (11/12), increasing from 61% to 75%.

During the training, all trainers were tasked with developing an action plan to outline how they would disseminate what they had learnt and how they would cascade their training in their areas of work. In order to support this, a community of practice, WhatsApp group was developed helping them to stay connected and share their learning, challenges and questions they may have when cascading the training. Additionally, each trainer was provided with a K50 (Aus$20) phone card every month for three months to purchase data, allowing them to access the internet, assist others in downloading the app and cascade the training effectively.

To date, further training has been conducted by 10 of the initial trainers, with skills-based training using the App reaching 52 second and final year student nurses. One school of nursing plans to integrate the App into lesson plans for second- and third-year student nurses. An orientation to the App has been conducted for 21 clinical staff in two hospitals; and a 2-day training has taken place for all nurse educators at the school of nursing. One trainer, who works very remotely, has provided training to five of her staff in the more remote areas, with support from two of the nursing educators/ safe delivery app trainers. Training in use of the App has been included in regular in-service training; and one trainer facilitated skills stations at the annual PNG Midwifery Symposium, held in Port Moresby in November 2024.

## Modules of the Safe Delivery App (PNG version)

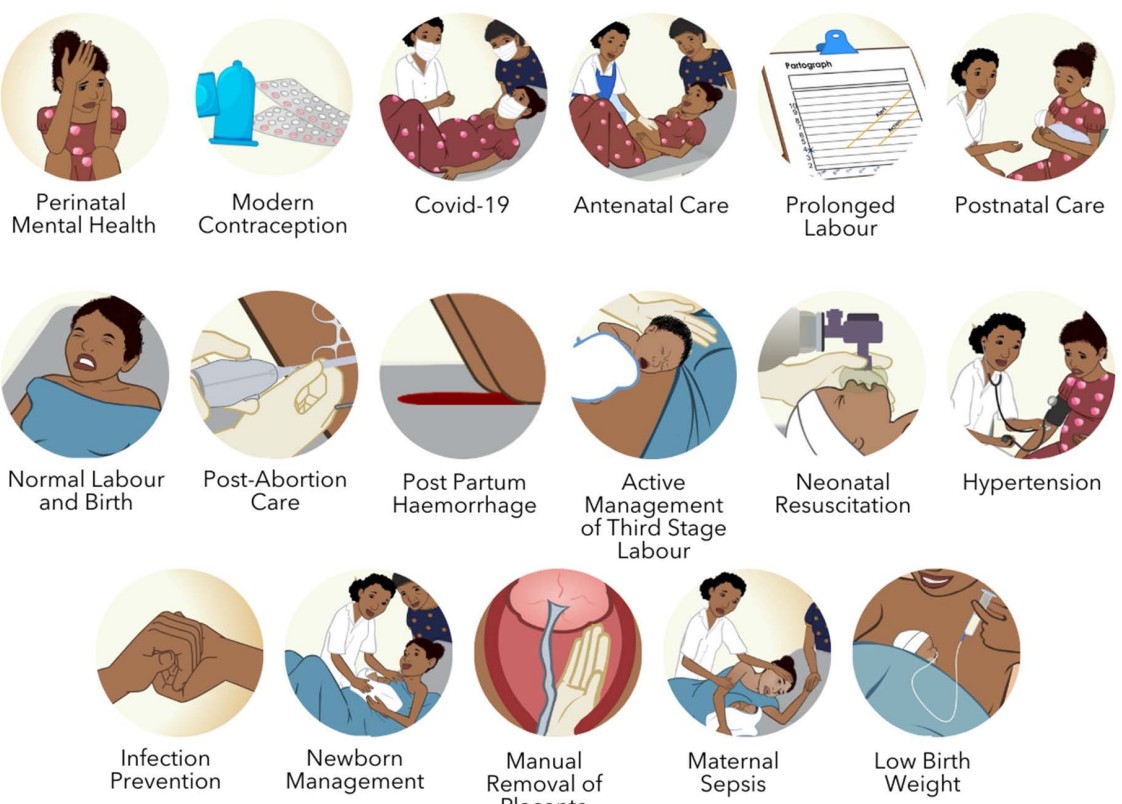

**Fig 5. Modules available in the PNG version of the App.**

## Discussion

The Safe Delivery App was implemented, adapted and scaled up in PNG between 2022–2024. There was an increase in both downloads of the App and registered users over the study period. Most registered users were midwives and nurses (78%). Overall, the App has been shown to be acceptable, being used in clinical settings, as a reference tool, as well as to support emergency situations and in nursing and midwifery education.

The App was seen as an appropriate tool to provide up to date evidence-based care in a country with a limited number of midwives, resulting in nurses frequently managing women and newborns with little support. The importance of the App being applicable to the PNG context, aligned with the national clinical guidelines, and supported by senior clinicians in the country was seen as paramount. While many nurses and midwives referred to the "Red Book" when discussing clinical care of pregnant women and newborn babies, in reality, the book is often not available. Student midwives receive a copy during their training, but nurses frequently are not provided a copy, and they can be difficult to access, particularly in the remote areas. Adherence to evidence based clinical guidelines can lead to improved quality of care [20], and in the absence of the "Red Book", the App provides on-the-spot step by step instructions to support clinical care.

Most of the participants in our study had received some training in the use of the App. Among those who had not received formal training or instruction in the use of the App, it was found to be easy to use. This is similar to the

experience in other countries. For example, a study from India identified that health care workers who had received instruction and facilitation in downloading and using the App had better retention of knowledge and confidence when dealing with emergency situations [11]. Orientation into the use of the App, including all five features is also linked to an increase in the use of the MyLearning platform and subsequently to users becoming Safe Delivery Champions [11].

Among participants attending the training of trainers, confidence and practical skills in managing prevention and management of bleeding after birth increased. Similarly, from Ethiopia, Thomsen et al [7] report that nurse-midwives and health extension workers afraid of complications because of a lack of skills, felt less afraid and more confident in their practice, following their access to the App. However, they went on to note that although the App appears to empower health workers and give them confidence, they may also be left feeling exposed or frustrated if particular supplies and equipment are not available [7].

The critical shortage of all cadres of health care workers in PNG, particularly midwives [15], leaves many, particularly in the rural areas facing challenges in the provision of emergency maternal and newborn care. In addition, there is a lack of ongoing continuing professional development or access to regular education on emergency, especially up-to-date information for many health care workers in PNG [16]. The use of mobile phones as a tool for health workers to collect data [21,22], and to communicate with clinicians when a higher level of care is needed [23], is reported from Ethiopia, Uganda and PNG [24]. A few studies have focused on the use of mobile phones to improve knowledge and skills of health workers working with maternal and newborn care in LMIC settings [6,7,25].

Of the five features included in the App, the video chapters were seen as providing practical, clear instructions, helping users translate their theoretical knowledge into practice. When working through the practical, skills-based activities participants in our training of trainers cohort referred to the appropriate video chapters, talking each other through the process and practice to ensure all steps were completed. Participants reported that they had used video clips while managing cases; and nurse educators reported using the App when training student midwives in clinics. The use of videos to support training are reported to effective from elsewhere [25]. These findings are similar to those reported from a study among nurses and midwives in India with the videos as the most frequently used feature [11]. In addition, mHealth tools, such as the Safe Delivery App, providing visual information and instruction is an opportunity to improve knowledge and skills of health workers, particularly those serving in remote areas [25].

Interestingly, the download data from PNG identified that the most frequently accessed feature was "action cards" followed by "MyLearning" and "Video chapters". This could be because the App is being used for training purposes, with the MyLearning used to guide knowledge gain. In India, the MyLearning feature was the most accessed of the five features, however the training in the use of the App was included within a wider training program, mandating that all those included in the training must become Safe Delivery Champions [26], which may be why the feature was accessed so frequently.

Midwifery educators are essential to strengthening the midwifery workforce in PNG and other LMICs. Updating skills and knowledge, to ensure that women receive quality midwifery care, is critical to building the capacity among nurse and midwifery educators [27]. The uptake and use of the App within nursing and midwifery schools, and the increased use among student nurses and midwives provides the opportunity for evidence-based care to be provided even in the more remote areas of PNG. However, there is a need to identify barriers to those not accessing the App to support all health care workers, particularly in the more remote areas [28]. Dissemination of the App has been ongoing throughout the study period however there remains a need to identify how to reach nurses and midwives working clinically and remotely if we are to improve quality of care through evidence-based practice.

## Conclusion

The Safe Delivery App is seen as a useful tool, supporting clinical practice, knowledge and skills, providing users with more confidence in their ability to provide quality maternal and newborn health care. Wider implementation of the App

across PNG may be a potential way to support all health care workers in the remote settings, providing on the spot, up to date evidence based clinical guidance.

## Supporting information

**S1 File. Coding framework _ Key informant summary data.**
(DOCX)

## Acknowledgments

We would like to acknowledge the support of workshop participants who helped in refining the clinical guidelines, ensuring the alignment to the PNG national guidelines: Dr Mary Bagita, Dr Robert Jones, Dr Francesca Failing, Dr Gamini Vali, Dr. Roland Barnabas, Sr Mary Sitaing, Sr Ellie Korave, Sr Julie Api, Sr Annette Semo, Sr Mary Anne Maga.

## Author contributions

**Conceptualization:** Delly Babona, Caroline S. E. Homer, Lisa Michelle Vallely.

**Data curation:** Lucy Au, Cherolyn Polomon, Arpita Deb, Disha Agarwalla, Lisa Michelle Vallely.

**Formal analysis:** Lisa Michelle Vallely.

**Funding acquisition:** Delly Babona, Caroline S. E. Homer.

**Methodology:** Delly Babona, Lisa Michelle Vallely.

**Project administration:** Delly Babona.

**Resources:** Arpita Deb, Disha Agarwalla, Michaela A. Riddell, Angela Kelly-Hanku.

**Supervision:** Delly Babona, Lisa Michelle Vallely.

**Validation:** Lisa Michelle Vallely.

**Writing – original draft:** Lisa Michelle Vallely.

**Writing – review & editing:** Delly Babona, Lucy Au, Cherolyn Polomon, Arpita Deb, Hilde Cortier, John Bolnga, Michaela A. Riddell, Angela Kelly-Hanku, Caroline S. E. Homer.

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
