## [Decision Letter · Decision Letter 0]

21 Jun 2025

Dear Dr. Vallely,

Thank you for submitting your manuscript to PLOS ONE. After careful consideration, we feel that it has merit but does not fully meet PLOS ONE’s publication criteria as it currently stands. Two reviewers have provided constructive feedback below. We invite you to submit a revised version of the manuscript that addresses the points raised during the review process.

We look forward to receiving your revised manuscript.

Kind regards,

Hannah Tappis, DrPH, MPH

Academic Editor

PLOS ONE

**Journal Requirements:**

1. When submitting your revision, we need you to address these additional requirements. Please ensure that your manuscript meets PLOS ONE's style requirements, including those for file naming. The PLOS ONE style templates can be found at https://journals.plos.org/plosone/s/file?id=wjVg/PLOSOne_formatting_sample_main_body.pdf and https://journals.plos.org/plosone/s/file?id=ba62/PLOSOne_formatting_sample_title_authors_affiliations.pdf 2. Please include a complete copy of PLOS’ questionnaire on inclusivity in global research in your revised manuscript. Our policy for research in this area aims to improve transparency in the reporting of research performed outside of researchers’ own country or community. The policy applies to researchers who have travelled to a different country to conduct research, research with Indigenous populations or their lands, and research on cultural artefacts. The questionnaire can also be requested at the journal’s discretion for any other submissions, even if these conditions are not met. Please find more information on the policy and a link to download a blank copy of the questionnaire here: https://journals.plos.org/plosone/s/best-practices-in-research-reporting. Please upload a completed version of your questionnaire as Supporting Information when you resubmit your manuscript. 3. Thank you for stating the following financial disclosure: This study was made possible through philanthropic donations from the Finkel Foundation and a Burnet Institute Christmas Appeal   Please state what role the funders took in the study.  If the funders had no role, please state: "The funders had no role in study design, data collection and analysis, decision to publish, or preparation of the manuscript." If this statement is not correct you must amend it as needed. Please include this amended Role of Funder statement in your cover letter; we will change the online submission form on your behalf. 4. Thank you for stating the following in the Acknowledgments Section of your manuscript: We would like to acknowledge the support of workshop participants who helped in refining the clinical guidelines, ensuring the alignment to the PNG national guidelines: Dr Mary Bagita, Dr Robert Jones, Dr Francesca Failing, Dr Gamini Vali, Dr. Roland Barnabas, Sr Mary Sitaing, Sr Ellie Korave, Sr Julie Api, Sr Annette Semo, Sr Mary Anne Maga. This study was made possible through philanthropic donations from the Finkel Foundation and a Burnet Institute Christmas Appeal We note that you have provided funding information that is not currently declared in your Funding Statement. However, funding information should not appear in the Acknowledgments section or other areas of your manuscript. We will only publish funding information present in the Funding Statement section of the online submission form. Please remove any funding-related text from the manuscript and let us know how you would like to update your Funding Statement. Currently, your Funding Statement reads as follows: This study was made possible through philanthropic donations from the Finkel Foundation and a Burnet Institute Christmas Appeal  Please include your amended statements within your cover letter; we will change the online submission form on your behalf. 5. In the online submission form, you indicated that “Qualitative data are available from the PI for the study, on reasonable request lvallely@kirby.unsw.edu.au”.  All PLOS journals now require all data underlying the findings described in their manuscript to be freely available to other researchers, either a. In a public repository, b. Within the manuscript itself, or c. Uploaded as supplementary information.This policy applies to all data except where public deposition would breach compliance with the protocol approved by your research ethics board. If your data cannot be made publicly available for ethical or legal reasons (e.g., public availability would compromise patient privacy), please explain your reasons on resubmission and your exemption request will be escalated for approval. 6. Please amend either the abstract on the online submission form (via Edit Submission) or the abstract in the manuscript so that they are identical.

**Additional Editor Comments: **

Please ensure that the manuscript adheres to reporting standards applicable to the study design, available at https://www.equator-network.org/reporting-guidelines/

Reviewers' comments:

Reviewer's Responses to Questions

**Comments to the Author**

1. Is the manuscript technically sound, and do the data support the conclusions?

Reviewer #1: Yes

Reviewer #2: Yes

2. Has the statistical analysis been performed appropriately and rigorously?

Reviewer #1: N/A

Reviewer #2: Yes

3. Have the authors made all data underlying the findings in their manuscript fully available?

Reviewer #1: No

Reviewer #2: No

4. Is the manuscript presented in an intelligible fashion and written in standard English?

Reviewer #1: Yes

Reviewer #2: Yes

**Reviewer #1:**  Please describe the pre-and-post test noted in line 424 in the methodology section - including questionnaire developing and pre-testing etc - and analysis used to achieve these results.

The paper notes adaptation of the app to local guidelines but doesn't describe if PNG guidelines are up to date in alignment with global standards - please add language to discuss this as - in theory - the App is in alignment with global guidelines

Please source other papers that have reported on testing the App in other contexts (Rwanda, DR Congo etc)

**Reviewer #2: ** General:

Interesting to see the continued dissemination of the App through both formal and informal mechanisms after the initial training and launch activities were complete - one can infer continued interest within networks of health care workers.

The Data Availability Statement should address where the data can be accessed without restriction, in line with the requirements of the section for publication.

Would edit out use of passive voice throughout.

Specific suggested edits:

Line 39 - would suggest "adapt" rather than "develop".

Lines 212 - 213: For reader understanding it would be good to clarify how downloads reached 1873 in Dec 2024 but 1495 downloads are reported from 2022 - 2024; if baseline started at 606, 1495 downloads would still be higher than 1873. Does this figure take App deletions into consideration or is there another explanation?

Lines 225 - 226: This reads a bit differently than what was reported in lines 121 - 123 "Half of the users had learnt about the App through in-service training, others had received stand-alone training in the use of the App, learnt about it through a colleague or friend or through a conference or other event."

Lines 278 - 279: This is a promising outcome of utilization of the App. Is this qualitatively reported by the HCWs (and if so, would add "reportedly"), or something that was documented and can be confirmed?

Line 358 (section): in the challenges, was there any information provided by the HCWs interviewed regarding supplies? While beyond the scope of the Safe Delivery App, it would effect if and how the HCWs could put the directives to use. For example, if the App explains up to date guidance of management of PPH which includes certain drugs, medications, and commodities which may not be available - how did they overcome this, or was this addressed in the adaptation? If not it's also ok because this is touched on in the discussion, but would be a good addition to results if this was covered.

Lines 424 - 425: Unsure if skills should be mentioned as there is no clear explanation on how this was determined or score increase on skills test provided. Suggest removing, or adding more supporting data.

Line 459: Are the PNG version of the App and the Red Book fully aligned and contain the same content? If so would be strong to emphasize that.

518 - 520: The last line is a bit confusing, it makes it seem as if all of the midwives, nurses, other HCWs that were spoken about are "unskilled" or that the App is for non-skilled providers. Authors may be trying to say where staffing resources are low and HCWs have to deal with increased case load, manage complications, or where continuing professional development/ refresher training is rare?

**Do you want your identity to be public for this peer review?** For information about this choice, including consent withdrawal, please see our Privacy Policy

Reviewer #1: **Yes: ** Nancy Bolan

Reviewer #2: **Yes: ** Lauren Bellhouse

---

## [Author Response · Author response to Decision Letter 1]

16 Jul 2025

Editorial comments

1. When submitting your revision, we need you to address these additional requirements. Please ensure that your manuscript meets PLOS ONE's style requirements, including those for file naming.

Many thanks I have reviewed and edited as required.

2. Please include a complete copy of PLOS’ questionnaire on inclusivity in global research in your revised manuscript.

Many thanks for this.

We have now uploaded a completed questionnaire.

This study was made possible through philanthropic donations from the Finkel Foundation and a Burnet Institute Christmas Appeal

Please state what role the funders took "The funders had no role in study design, data collection and analysis, decision to publish, or preparation of the manuscript." in the study. If the funders had no role, please state:

This has been included in the revised cover letter

We would like to acknowledge the support of workshop participants who helped in refining the clinical guidelines, ensuring the alignment to the PNG national guidelines: Dr Mary Bagita, Dr Robert Jones, Dr Francesca Failing, Dr Gamini Vali, Dr. Roland Barnabas, Sr Mary Sitaing, Sr Ellie Korave, Sr Julie Api, Sr Annette Semo, Sr Mary Anne Maga. This study was made possible through philanthropic donations from the Finkel Foundation and a Burnet Institute Christmas Appeal

This study was made possible through philanthropic donations from the Finkel Foundation and a Burnet Institute Christmas Appeal

The following statement has now been removed from the Acknowledgements section of the manuscript.

“This study was made possible through philanthropic donations from the Finkel Foundation and a Burnet Institute Christmas Appeal”.

I had wanted to add this to the Funding information section but was unable to identify how to include such a donation, without Award number etc.

The following statement should be included

“This study was made possible through philanthropic donations from the Finkel Foundation and a Burnet Institute Christmas Appeal”

5. In the online submission form, you indicated that “Qualitative data are available from the PI for the study, on reasonable request lvallely@kirby.unsw.edu.au”.

This policy applies to all data except where public deposition would breach compliance with the protocol approved by your research ethics board.

If your data cannot be made publicly available for ethical or legal reasons (e.g., public availability would compromise patient privacy), please explain your reasons on resubmission and your exemption request will be escalated for approval.

I have uploaded the code book as supplementary information (Supp File 2) that we developed to draw our data and conclusions from.

I have amended the Data availability section, including the following

The code book for the qualitative data is included as a supplementary file. Additional requests for qualitative data may be forwarded to info@pngimr.org.pg

Relating to the user data, please see response from the Maternity Foundation:

Safe Delivery App user data: The data used in this study were collected through the Safe Delivery App and stored on cloud servers hosted by the Maternity Foundation in India and Central Europe. Due to privacy concerns and user confidentiality agreements, the raw user-level data cannot be shared publicly. However, aggregated and anonymised summary statistics that support the findings of this study are available from the Maternity Foundation upon reasonable request. Researchers seeking access may contact the Maternity Foundation at mail@maternity.dk and must comply with applicable ethical and data privacy regulations.

6. Please amend either the abstract on the online submission form (via Edit Submission) or the abstract in the manuscript so that they are identical.

Abstract in manuscript aligned with that on the online submission.

7. Please review your reference list to ensure that it is complete and correct. If you have cited papers that have been retracted, please include the rationale for doing so in the manuscript text or remove these references and replace them with relevant current references.

Any changes to the reference list should be mentioned in the rebuttal letter that accompanies your revised manuscript.

If you need to cite a retracted article, indicate the article’s retracted status in the References list and also include a citation and full reference for the retraction notice.

Many thanks, reviewed and correct

I have included the added references the cover letter

Please ensure that the manuscript adheres to reporting standards applicable to the study design, available at https://www.equator-network.org/reporting-guidelines/ Many thanks, reviewed.

Comments from reviewers

1. Is the manuscript technically sound, and do the data support the conclusions?

Reviewer #1: Yes

Reviewer #2: Yes No comment required

2. Has the statistical analysis been performed appropriately and rigorously?

Reviewer #1: N/A

Reviewer #2: Yes No comment required

3. Have the authors made all data underlying the findings in their manuscript fully available?

The PLOS Data policy requires authors to make all data underlying the findings described in their manuscript fully available without restriction, with rare exception (please refer to the Data Availability Statement in the manuscript PDF file).

The data should be provided as part of the manuscript or its supporting information or deposited to a public repository. For example, in addition to summary statistics, the data points behind means, medians and variance measures should be available. If there are restrictions on publicly sharing data—e.g. participant privacy or use of data from a third party—those must be specified.

Reviewer #1: No

Reviewer #2: No

Please see point # 4 above.

4. Is the manuscript presented in an intelligible fashion and written in standard English?

Reviewer #1: Yes

Reviewer #2: Yes No comment required

Reviewer #1

Please describe the pre-and-post-test noted in line 424 in the methodology section - including questionnaire developing and pre-testing etc - and analysis used to achieve these results.

The pre- and post-test used in this study follows the standardised pre- and post-test developed by clinical training experts at Maternity Foundation. The test content aligns with the contents of the Safe Delivery App and was initially tested in projects in Ethiopia. It has since been adapted and used in several other countries.

The paper notes adaptation of the app to local guidelines but doesn't describe if PNG guidelines are up to date in alignment with global standards - please add language to discuss this as - in theory - the App is in alignment with global guidelines

Please source other papers that have reported on testing the App in other contexts (Rwanda, DR Congo etc)

Many thanks for highlighting this, I have added to the sentence on P6, line 183 to provide clarity.

It now reads:

Clinical content was reviewed and adapted to the local context ensuring alignment with the current Obstetrics and Gynaecology and Paediatric clinical Guidelines.(15, 16)

Many thanks for this most recent work, I have included in the introduction section of the manuscript the following references

Ref #9

Nishimwe A, Conco DN, Nyssen M, Ibisomi L. A mixed-method study exploring experiences, perceptions, and acceptability of using a safe delivery mHealth application in two district hospitals in Rwanda. BMC Nurs. 2022;21(1):176.

Ref #10

Oladeji O, Tessema M, B O. Strengthening quality of maternal and newborn care using catchment based clinical mentorship and safe delivery app: A case study from Somali region of Ethiopia. Int J Midwifery and Nursing Practice. 2022;5(1):13-8.

Ref # 11

Sarin E, Dastidar SG, Bisht N, Bajpayee D, Patel R, Sodha TS, et al. Safe Delivery application with facilitation increases knowledge and confidence of obstetric and neonatal care among frontline health workers in India. J Family Med Prim Care. 2022;11(6):2695-708

Reviewer #2: General

Interesting to see the continued dissemination of the App through both formal and informal mechanisms after the initial training and launch activities were complete - one can infer continued interest within networks of health care workers.

The Data Availability Statement should address where the data can be accessed without restriction, in line with the requirements of the section for publication.

Would edit out use of passive voice throughout.

Please see response to Editorial comment 3 above.

Line 39 - would suggest "adapt" rather than "develop".

Mant thanks, altered in manuscript

P2, line 39

“In this paper we describe the uptake and acceptability and the process to align the App in Papua New Guinea (PNG)”.

Lines 212 - 213: For reader understanding it would be good to clarify how downloads reached 1873 in Dec 2024 but 1495 downloads are reported from 2022 - 2024; if baseline started at 606, 1495 downloads would still be higher than 1873. Does this figure take App deletions into consideration or is there another explanation?

Many thanks for this comment. I have revised the sentence to make it clearer.

It now reads as ;

P7, lines 213-217

At the start of the study period there had been 378 downloads of the App. Between June 2022 and December 2024 there were an additional 1495 downloads. (Fig2); 87% of users (1304/1495) had registered profiles, , providing detail about use of the App.

Lines 225 - 226: This reads a bit differently than what was reported in lines 121 - 123 "Half of the users had learnt about the App through in-service training, others had received stand-alone training in the use of the App, learnt about it through a colleague or friend or through a conference or other event." Many thanks for this comment. The difference between these two statements is that the earlier lines (121-123) are from the earlier, hence why they are included in the introduction.

In lines 2225-226 we are referring to the findings from our work during the study period.

In considering this, no changes have been made to the manuscript.

Lines 278 - 279: This is a promising outcome of utilization of the App. Is this qualitatively reported by the HCWs (and if so, would add "reportedly"), or something that was documented and can be confirmed?

Many thanks for this comment. We have altered this paragraph for clarity.

It now reads;

P 9, lines 279-281

The App has enabled health care workers in the health centres to manage situations without the need to refer women to hospital, with one participant reporting a reduced number of referrals. Another participant reported that after training, staff were able to use the App, leading to a reduced need to call support from senior staff.

Line 358 (section): in the challenges, was there any information provided by the HCWs interviewed regarding supplies? While beyond the scope of the Safe Delivery App, it would effect if and how the HCWs could put the directives to use. For example, if the App explains up to date guidance of management of PPH which includes certain drugs, medications, and commodities which may not be available - how did they overcome this, or was this addressed in the adaptation? If not it's also ok because this is touched on in the discussion, but would be a good addition to results if this was covered.

Many thanks for this comment. While there was no mention of commodities not being available, this may be a re flection of the health care workers we interviewed. It will be important to consider this in future evaluation of the App in PNG.

Lines 424 - 425: Unsure if skills should be mentioned as there is no clear explanation on how this was determined or score increase on skills test provided. Suggest removing, or adding more supporting data.

Many thanks for highlighting this. On reflection, we do not have data to support the line on improved skills, and we have removed the line completely

Line 459: Are the PNG version of the App and the Red Book fully aligned and contain the same content? If so would be strong to emphasize that.

Many thanks for this comment. I have added m more detail in the results section, lines 331-332 to emphasize the point that the Red Book is the clinical guidelines:

“A number of participants spoke of referring to national standard clinical guidelines for obstetric care, often referring to “the red book”, a small, pocket sized book outlining key aspects of care during pregnancy and childbirth, aligned with the national standards.”

518 - 520: The last line is a bit confusing, it makes it seem as if all of the midwives, nurses, other HCWs that were spoken about are "unskilled" or that the App is for non-skilled providers. Authors may be trying to say where staffing resources are low and HCWs have to deal with increased case load, manage complications, or where continuing professional development/ refresher training is rare?

Many thanks. I have amended to read the following:

“…to support all health care workers in the remote settings, providing on the spot, up to date evidence based clinical guidance.”

6. PLOS authors have the option to publish the peer review history of their article (what does this mean?). If published, this will include your full peer review and any attached files.

The authors consent to the publication of the peer review history of this article.

---

## [Editor Report · Decision Letter 1]

22 Jul 2025

"It's like a book in the palm of my hand": Adapting the Safe Delivery App for Papua New Guinea to improve quality of maternal and newborn care

PONE-D-25-20687R1

Dear Dr. Vallely,

We’re pleased to inform you that your manuscript has been judged scientifically suitable for publication and will be formally accepted for publication once it meets all outstanding technical requirements.

Kind regards,

Hannah Tappis, DrPH, MPH

Academic Editor

PLOS ONE

Additional Editor Comments (optional): Previous reviewer comments have been thoughtfully addressed. 